# Endoscopist’s Judgment Is as Useful as Risk Scores for Predicting Outcome in Peptic Ulcer Bleeding: A Multicenter Study

**DOI:** 10.3390/jcm9020408

**Published:** 2020-02-03

**Authors:** Enric Brullet, Pilar Garcia-Iglesias, Xavier Calvet, Michel Papo, Montserrat Planella, Albert Pardo, Félix Junquera, Silvia Montoliu, Raquel Ballester, Eva Martinez-Bauer, David Suarez, Rafel Campo

**Affiliations:** 1Hospital de Sabadell, Corporació Sanitària Universitària Parc Taulí, 08208 Sabadell, Spain; ebrullet@tauli.cat (E.B.); pgarciai@tauli.cat (P.G.-I.); fjunquera@tauli.cat (F.J.); emartinezb@tauli.cat (E.M.-B.);; 2Departament de Medicina, Universitat Autònoma de Barcelona, 08208 Sabadell, Spain; 3CIBEREHD—Instituto de Salud Carlos III, 28029 Madrid, Spain; 4Hospital Joan XXIII, 43005 Tarragona, Spain; mpapo.hj23.ics@gencat.cat (M.P.); alberto.pardo@urv.cat (A.P.); smontoliu.hj23.ics@gencat.cat (S.M.); 5Hospital Universitari Arnau de Vilanova, 25198 Lleida, Spain; mplanella.lleida.ics@gencat.cat (M.P.); rballesterclau@gmail.com (R.B.); 6Unitat d’Epidemiologia i Avaluació, Fundació Parc Taulí, Universitat Autónoma de Barcelona, 08208 Sabadell, Spain; david.suarez.lamas@gmail.com

**Keywords:** peptic ulcer bleeding, risk score, endoscopist’s judgement

## Abstract

**Background:** Guidelines recommend using prognostic scales for risk stratification in patients with non-variceal upper gastrointestinal bleeding. It remains unclear whether risk scores offer greater accuracy than clinical evaluation. **Objective:** Compare the diagnostic accuracy of the endoscopist’s judgment against different risk-scoring systems (Rockall, Glasgow–Blatchford, Baylor and the Cedars–Sinai scores) for predicting outcomes in peptic ulcer bleeding (PUB). **Methods:** Between February 2006 and April 2010 we prospectively recruited 401 patients with peptic ulcer bleeding; 225 received endoscopic treatment. The endoscopist recorded his/her subjective assessment (“endoscopist judgment”) of the risk of rebleeding and death immediately after endoscopy for each patient. Independent evaluators calculated the different scores. Area under the receiver-operating-characteristics (ROC) curve, sensitivity, specificity, positive and negative predictive values were calculated for rebleeding and mortality. **Results:** The areas under ROC curve of the endoscopist’s clinical judgment for rebleeding (0.67–0.75) and mortality (0.84–0.9) were similar or even superior to the different risk scores in both the whole group and in patients receiving endoscopic therapy. **Conclusions:** The accuracy of the currently available risk scores for predicting rebleeding and mortality in PUB patients was moderate and not superior to the endoscopist’s judgment. More precise prognostic scales are needed.

## 1. Introduction

Non-variceal upper gastrointestinal bleeding (NVUGB) is a common reason for hospital admission and a major cause of mortality, morbidity and health-care expenditure [1,2]. Peptic ulcer bleeding (PUB) is its most frequent etiology. In recent years, improvements in the management of PUB have reduced the risk of rebleeding and death. Specifically, in patients with most severe bleeding—those with high-risk endoscopic stigmata—the combination of aggressive volume repletion and rapid correction of hypotension [3], endoscopic therapy and intravenous proton-pump inhibitors (PPI) has been shown to improve outcomes [4,5,6,7,8,9,10].

Patients with PUB present a wide range of clinical severity, from minor bleeding to fatality. Several scoring systems have been designed to identify patients with high risk of adverse outcomes (commonly defined as a risk of rebleeding over 5% and a mortality over 1%) and to differentiate them from lower-risk patients [11,12,13,14].

Management of an individual patient depends on the accurate estimation of the risk for rebleeding and death. The international consensus on NVUGB recommends the stratification of patients into low and high risk by using prognostic scales [6,15,16]. Early identification of high-risk patients may allow appropriate intervention, thus minimizing morbidity and mortality. However, even though international recommendations strongly support their use, risk scores are not widely applied in clinical practice [1].

Over the last few decades, many risk scores combining clinical, laboratory or/and endoscopic parameters, have been developed with the aim of assisting physicians in the early stages of decision making. Several prognostic scores have been developed, in order to identify high-risk patients, who require immediate treatment, and patients at low risk whom endoscopy may be delayed. The quality of the development process and the degree of further clinical validation of the different scores has been variable. The aim of these multiple scores has been also different (prediction of rebleeding, the need for intervention and mortality). Some tried to identify patients most likely to suffer an adverse outcome and, therefore, more likely to benefit from early, aggressive management whereas others attempted to identify those most likely to have a benign course so that early hospital discharge or even outpatient management can be considered.

Despite international recommendations, scores have not achieved widespread use in clinical practice. There are many possible reasons for this situation—among them, the lack of reference scores among the many scales that have been published and the fact that clinicians may not perceive that they offer a real benefit [1]. In fact, some studies suggest that clinical judgment may be as effective as most of the currently available risk scores for predicting the need for endoscopic therapy or other medical interventions in NVUGIB [17,18,19].

Risk scores have not been prospectively compared with clinical judgement (“clinical intuition”) of endoscopist to predict adverse outcome in patients with PUB. Therefore, the aim of the present study was to prospectively compare the diagnostic accuracy of the endoscopist’s judgment against different risk-scoring systems (Rockall, Glasgow–Blatchford, Baylor and the Cedars–Sinai scores) for predicting either mortality or rebleeding in PUB.

## 2. Methods

From February 2006 to April 2010, a prospective multicenter study was conducted at three tertiary-care university-affiliated hospitals in Catalonia (Spain). Inclusion criteria were: (1) upper gastrointestinal bleeding, defined by the presence of hematemesis, melena, hematochezia or blood from nasogastric tube aspiration; (2) peptic ulcer confirmed by upper gastrointestinal endoscopy (UGIE) performed within 24 h after the onset of bleeding; (3) age ≥ 18 years. Patients who developed PUB while hospitalized for an unrelated disease were also included. Four hundred and one consecutive patients with PUB were recruited; 225 of them received endoscopic treatment (Forrest I, IIa IIb). The Institutional Review Boards of the participating hospitals approved the study and informed consent was obtained from all subjects. 

The study was performed in accordance with the Standards for Reporting of Diagnostic Accuracy Studies (STARD) recommendations for performing diagnostic accuracy studies [20].

### 2.1. Endoscopy 

At admission, patients received standard clinical care including blood volume restoration and intravenous PPI. UGIE was performed within 6–24 h after admission by the on-call endoscopist. Twenty-five different endoscopists performed the procedures; all had five or more years of experience in therapeutic endoscopy. Patients with spurting hemorrhage (Forrest Ia), oozing bleeding (Forrest Ib), non-bleeding visible vessel (NBVV; Forrest IIa), and adherent clots (Forrest IIb) received endoscopic treatment. Endoscopic therapy consisted of injection of epinephrine followed by either polidocanol or clipping, or clipping alone, according to the endoscopist’s preferences. 

### 2.2. Clinical Management and Outcomes

After endoscopic treatment, all patients were treated with high-dose PPI in accordance with current recommendations [6]. Permanent hemostasis was defined as successful initial hemostasis without the development of rebleeding. Rebleeding was suspected when a patient developed fresh hematemesis, hypotension (systolic blood pressure <100 mm Hg) or tachycardia (pulse >100 beat/min), or persistent melena or required a total transfusion of four units or more to maintain hemoglobin level. In all patients, rebleeding was confirmed by endoscopy. A second endoscopic treatment was applied whenever possible. Patients with persistent bleeding after a second endoscopic treatment were treated either with surgery or arterial embolization, depending on the clinical situation and local expertise. Primary outcomes were 30-day rebleeding and mortality. Gold standard regarding these outcomes was stablished according to patients’ clinical follow-up.

### 2.3. Data Collection

Following UGIE, the endoscopist recorded his/her subjective assessment (“endoscopist judgment”) of each patient’s risk of rebleeding according to the clinical report and endoscopic findings. Endoscopist classified the patients in low and high-risk groups for both rebleeding and mortality. No guideline was provided to the endoscopists on how to evaluate the risk. In an attempt to prevent the endoscopist from inadvertently using the scores, the values of the variables necessary for calculating the different scoring systems were obtained after the endoscopist’s evaluation by a gastroenterologist who was not involved in the initial evaluation and was blind to the endoscopist’s opinion. 

Risk scores were calculated and graded according to the criteria reported by the authors of the original articles in which the scoring systems were introduced [11,12,13,14]. Rockall scoring system values range from 0 to 11. Risk categories are high (≥5), intermediate (3–4), and low (0–2). Cedars–Sinai scores range from 0 to 11, and risk is classified as high (≥5), intermediate (3–4), and low (0–2). Glasgow–Blatchford scores range from 0 to 23; any score higher than 0 represents a high risk of needing medical intervention or transfusion. Finally, the Baylor score range is from 0 to 24, with high risk defined as >10. Cutoff values used in calculating sensitivity, specificity and positive and negative predictive values of high risk of rebleeding and mortality were >2 for the Rockall score, >2 for the Cedars–Sinai score, > 0 for the Glasgow–Blatchford score, and >10 for the Baylor score. This analysis was repeated using thresholds of the Rockall and Cedars–Sinai scores greater than 4 and the Glasgow–Blatchford score greater than 11 [11,12,13,14]. A 10% random sample of patients was rechecked to ensure the integrity of the data collection and the score calculations.

### 2.4. Statistical Analysis

Statistical analysis was performed with SPSS 21 (IBM, Chicago, IL, USA) and STATA. The discriminative ability of the scoring systems for predicting outcome was evaluated by the area under the receiver-operating-characteristics (ROC) curve with 95% confidence intervals. Areas under the ROC curve were calculated for the endoscopist’s judgment (mortality and rebleeding) and also for each of the risk scores. The different areas under the ROC curve were compared with each other and against the endoscopist’s judgment using STATA. A Bonferroni chi square test was used to correct for multiple comparisons, when comparing risk scores versus endoscopist’s judgment. Significance was set a *p* ≤ 0.05. A sample size of 400 patients was anticipated for an estimated rate of rebleeding of 10% a 10-percentage point-wide confidence interval and expecting a sensitivity of 80% and a specificity of 80%.

The predictive ability was further analyzed by calculating sensitivity, specificity, positive predictive values (PPV), and negative predictive values (NPV), for the Rockall, Cedars–Sinai, Glasgow–Blatchford and Baylor scores and endoscopist’s judgment (mortality and rebleeding). The cut-off values used to calculate these parameters where those previously described and conventionally accepted, as described above [11,12,13,14].

## 3. Results

During the study period, 401 patients presented with PUB; 225 showed endoscopic high-risk stigmata and received endoscopic therapy. 

Median age was 63.5 years (range 18–97), 37.7% of patients were aged over 70, and 70.6% were men. Sixty-one patients (15.2%) were hypotensive at the time of presentation (systolic blood pressure <100 mmHg or heart rate >100 per minute). The mean level of hemoglobin was 9.55 g/L ± 2.36 (range 3–17). Eleven patients (2.6%) needed surgery and five (1.1%) trans-catheter arterial embolization. 

Respective rebleeding and 30-day mortality rates were 10% (40 patients) and 3.2% (13 patients) for the whole group; the corresponding figures for the endoscopic treatment group were 16% (36 patients) and 4.9% (11 patients). Patients’ demographics and outcomes are shown in Table 1. 

Figure 1 and Figure 2 show the areas under the ROC curve for rebleeding and mortality for endoscopist’s clinical judgment and all the scores, both for the whole group and for the subgroup receiving endoscopic therapy. The endoscopist’s judgment achieved the highest area under the ROC curve in all settings.

### 3.1. Comparison of Endoscopists’ Judgement and Scores

In the whole group, areas under the ROC curve of the endoscopist judgment for mortality and rebleeding were 0.9 (95% confidence interval (CI): 0.864–0.937) and 0.753 (95% CI: 0.679–0.828) respectively. The comparison of the different areas under the ROC curve did not show significant differences in their predictive value for mortality (*p* = 0.0532). Nor did the Bonferroni chi square test comparing the different scores between them or versus the endoscopist’s judgment find significant differences (Figure 1). 

Regarding rebleeding, no significant differences were found between areas under ROC curves either in the global comparison (*p* = 0.0659) or when comparing the scores between them or with endoscopist’s judgment (Figure 1).

Areas under the ROC curve values for the subgroup of patients receiving endoscopic therapy were 0.840 (95% CI: 0.781–0.899) and 0.675 (95% CI: 0.588–0.763) for mortality and rebleeding respectively. Again, no significant differences were found either in the global comparison (*p* = 0.0674) or when comparing scores with the endoscopist’s judgment (Figure 2). Predictive values of both the endoscopists’s judgment and risk scores for the risk of rebleeding after endoscopic treatment were low to moderate. 

Regarding prediction of death, scoring systems and the endoscopist’s judgment both showed high sensitivity and negative predictive values, but low specificity and positive predictive values. For predicting rebleeding, risk scores and endoscopist’s judgment had acceptable sensitivity and negative predictive values, but again low specificity and positive predictive values (Table 2, Table 3, Table 4 and Table 5). 

### 3.2. Role of Endoscopist Experience

All endoscopists participating in the study were experienced, with experience of 8 to 21 years (mean 14.1 years) in endoscopy and at least 5 years in therapeutic endoscopy. In post-hoc analysis, endoscopists were empirically divided in two groups, more and less experienced taking into account whether they performed full-time or part-time endoscopy and the years of experience. No significant differences in the rates of adequate prediction were observed between the two groups. Sensitivity, specificity, positive and negative predictive values for both rebleeding and mortality were similar between the two groups (Table 6). 

## 4. Discussion 

The present study shows that scoring systems are not superior to the endoscopist’s clinical judgment for predicting either rebleeding or mortality in PUB. In addition, both endoscopist’s clinical judgment and scoring systems have only moderate reliability for predicting outcomes in PUB. This finding is striking, as it challenges the recent international consensus recommendations that risk scores should be used [6,15,16]. Furthermore, both clinical judgment and risk scores are least reliable in the setting in which they might be expected to be most useful: that is, in predicting rebleeding after endoscopic therapy. A reliable score in this setting would help guide decisions regarding the need for additional therapy and/or second look endoscopy in high-risk patients; at present, however, the predictive methods available are not sufficiently accurate to perform this role. 

A possible explanation for these rather unexpected results is that scores may not (or may only partially) detect difficult-to-measure variables. In this regard, the endoscopist’s subjective satisfaction with therapy, the presence of large vessels and the global burden of comorbidity are not well captured by current risk scores. In addition, substantial differences in the method of development, target population and outcome may contribute to the differences in reliability of the scores. A number of examples: (1) only the Baylor score was designed to predict rebleeding after endoscopic therapy [12]; the Rockall and Cedars–Sinai scores were designed to predict mortality [11,14], and the Glasgow–Blatchford score evaluates the need for further clinical intervention [13]. (2) The scores were calculated from different populations: Rockall, Cedars–Sinai and Glasgow–Blatchford scores [11,13,14] were derived from all patients with upper gastrointestinal bleeding, including variceal bleeding; only the Baylor score [12] was primarily derived from a selected population of patients with PUB who received endoscopic treatment. (3) The quality of the methods used to develop the scores and the degree of clinical validation is variable [11,12,13,14]; only 81.4% of the patients within the initial cohort studied to develop the Rockall score underwent upper endoscopy. Both Rockall and Cedars–Sinai scores included patients with bleeding due to portal hypertension and patients with bleeding and normal endoscopy. The Baylor score was developed from a small sample of 80 patients fulfilling the criteria for severe upper gastrointestinal bleeding (UGB). In addition, the results of the latter study were not externally validated.

In fact, and for whatever reason, risk scores are not widely used in clinical practice, despite being recommended by several international consensus documents. In the UK Comparative Audit of UGB of the British Society of Gastroenterology, only 19% (1250/6750) of cases had a risk score recorded in the medical notes (either Rockall or Glasgow Blatchford) [1]. This low compliance may reflect the perception that risk assessment by an experienced physician is often more accurate than anything the application of a mathematical model can provide. 

In a few studies, clinical judgment has been compared with scoring systems (mainly with the Glasgow–Blatchford score) to assess its ability to predict the need for additional medical interventions. Faroq et al. [17] demonstrated that the emergency department physicians’ decision to admit a bleeding patient to the ICU was an accurate predictor of the need for endoscopic therapy. They also showed that clinically triggered decisions were more accurate than the Rockall and Glasgow–Blatchford scores for detecting patients who presented rebleeding or developed adverse events. Attar et al. [18] compared the yield of the triage based on the Glasgow–Blatchford score with the endoscopist’s decision to perform urgent UGIE, and found that the score did not detect more patients needing urgent endoscopy than on-duty endoscopists. De Groot et al. [19] compared the “gut feeling” of the gastroenterologist with the Glasgow–Blatchford score for predicting the need for medical intervention in patients with upper gastrointestinal bleeding. They found that the Glasgow Blatchford score had a slightly better predictive area under ROC curve (0.85 vs. 0.77 respectively).

Our study also shows that scores perform differently for different outcomes and settings, and so it is not clear which score is the best for each situation. This result corroborates the few additional studies, which have compared the different risk scores in clinical practice. Assessing the usefulness of Rockall, Cedars–Sinai and Baylor scores in 343 patients with NVUGB, Camellini et al. [21] found that all three scores predicted mortality better than rebleeding. Moreover, areas under ROC curves for rebleeding were low, ranging from 0.59 to 0.67. Kim et al. [22] compared the clinical utility of scores for the prediction of rebleeding and death in 239 patients with NVUGB. The sensitivity of Rockall, Glasgow–Blatchford, Cedars–Sinai and Baylor scores in detecting the patients who died ranged from 100% to 87.5%, but the specificity was low (1.8%–58.5%). The sensitivity of Rockall, Cedars–Sinai and Baylor scores in predicting rebleeding was also low, ranging from 30.7% to 80%. The Glasgow–Blatchford score had a high sensitivity (94.2%) but an extremely low specificity (0.98%). A similar study from Taiwan compared the clinical utility of Rockall and Glasgow–Blatchford scores for the prediction of rebleeding and death in 303 patients with upper gastrointestinal bleeding; neither score performed well in predicting rebleeding and death [23]. Finally, Stanley et al. [24] compared the Rockall and Glasgow–Blatchford scores in the prediction of death in 1555 patients with UGB. Again, the area under the ROC curve for mortality was low for both scores. Large validation studies of the Rockall score have also found low areas under the ROC curve for rebleeding (<0.65). The authors concluded that the Rockall score was unsatisfactory for predicting rebleeding [25,26,27]. 

The strengths of the present study lie in its prospective and multicenter design and large sample size. However, the study has some limitations. First, the endoscopist’s clinical judgment (“clinical intuition”) is a subjective variable and is very difficult to standardize. Nonetheless, the reliability was high despite the large number of gastroenterologists evaluating the parameter. This suggests that clinical evaluation is quite straightforward and that neither extensive training nor long experience is necessary for determining patients’ prognosis. In addition, a post-hoc analysis in our study failed to show differences in the predictive value of clinical judgments depending on the endoscopist’s experience. Notably, however, all the endoscopists in the study had moderate to long experience. Therefore, it is possible that scores may be helpful for trainees or non-skilled practitioners. 

Since our study was performed, new scores, including AIMS65 have been published. AIMS65 was found to be most predictive of mortality in several studies comparing multiple bleeding risk scores [28,29]. Also, in a very recent study, a machine-learning model identified patients with upper gastrointestinal bleeding who met a composite endpoint of hospital-based intervention or death within 30 days with a great area under the curve (AUC) and high levels of specificity, at 100% sensitivity [30]. It would be very interesting to compare clinicians’ judgement with these new tools. 

In conclusion, endoscopist’s clinical judgment seems at least as effective as risk scores for predicting rebleeding and death after PUB. Furthermore, the four scores evaluated varied in terms of accuracy and performed better for the prediction of death than for the prediction of rebleeding. Overall, both the endoscopist’s clinical judgment and the scores had low specificities and positive predictive values and, therefore, a limited accuracy in predicting 30-day mortality and rebleeding. Our study casts doubt on the ability of scores to add relevant information to the more convenient, simple clinical evaluation performed by a gastroenterologist. Additional prospective studies are needed to confirm our data in different settings and to develop more accurate risk scores.

## Figures and Tables

**Figure 1 jcm-09-00408-f001:**
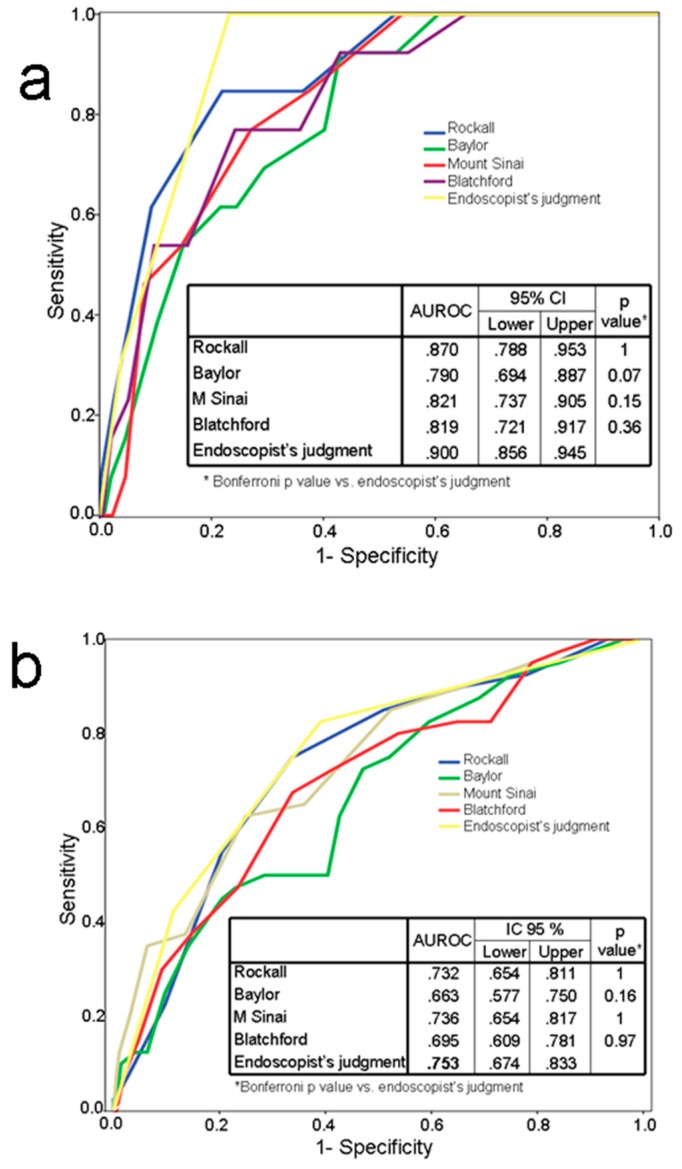
(**a**) Area under the receiver-operating-characteristics (ROC) curve for mortality. (**b**) Area under the ROC curve for rebleeding. All patients.

**Figure 2 jcm-09-00408-f002:**
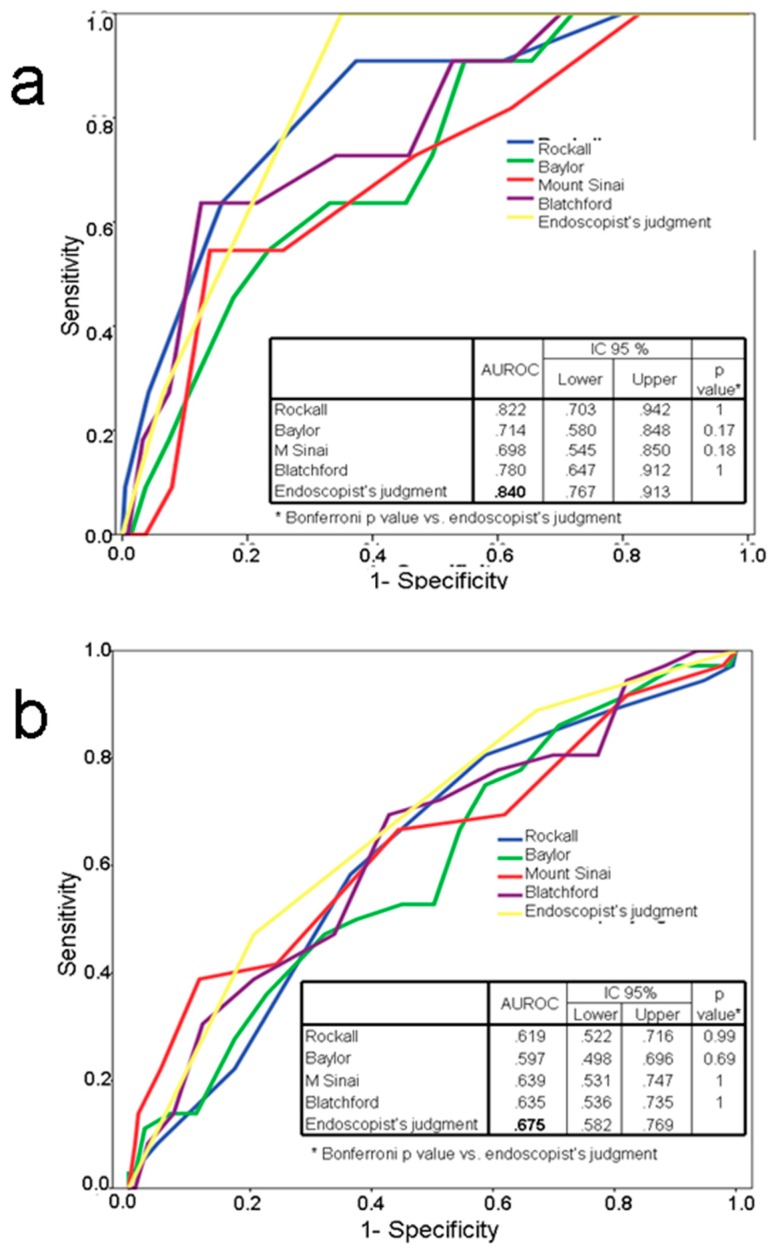
(**a**) Area under the ROC curve for mortality. (**b**) Area under the ROC curve for rebleeding. Endoscopically treated group.

**Table 1 jcm-09-00408-t001:** Demographic data and clinical characteristics in the study group.

Characteristics	Whole Group	Endoscopic Treatment
Number of patients	401 (100%)	225 (56.1%)
Mean age (years); *n* (SD)	63.5 (18.16)	63.5 (18.71)
Gender: Male	283 (71%)	154 (68%)
Ulcer site:		
gastric	162 (40%)	93 (41.3%)
duodenal	239 (60%)	132 (58.7%)
Forrest Classification:		
III (clean base)	81 (20%)	-
IIc (hematin flat spot)	95 (24%)	-
IIb (adherent blood clot)	53 (13%)	53 (24%)
IIa (non-bleeding visible vessel)	95 (24%)	95 (42%)
I (spurting/oozing)	77 (19%)	77 (34%)
Endoscopic treatment:		
Injection	132 (33%)	132 (58.7%)
Injection + clip	87 (22%)	87 (38.7%)
Clip	6 (2%)	6 (2.7%)
No treatment	176 (44%)	-
Hemoglobin (g/L); mean (SD)	9.55 (2.36)	9.20 (2.38)
Hypotension (%)	61 (15%)	59 (26%)
Blood transfusion	195 (48.6%)	134 (59.6%)
Surgery	11 (3%)	10 (4%)
Trans catheter embolization	5 (1%)	5 (2%)
Rebleeding	40 (10%)	26 (16%)
Mortality	13 (3%)	11 (5%)

**Table 2 jcm-09-00408-t002:** Sensitivity, specificity and positive predictive values for mortality (all patients).

	Sensitivity (%)	Specificity (%)	Positive Predictive Value (%)	Negative Predictive Value (%)
Endoscopist judgment	100	76.8	12.6	100
Rockall Score > 2	100	32.5	4.2	100
Rockall Score >4	84.6	63.7	7.2	99.2
Cedars–Sinai Score > 2	100	45.9	5.8	100
Cedars–Sinai Score >4	76.9	72.9	8.7	99.0
Blatchford Score >0	100	0.3	3.3	100
Blatchford Score >11	92.3	57.0	6.7	100
Baylor Score >10	69.2	70.6	7.3	98.6

**Table 3 jcm-09-00408-t003:** Sensitivity, specificity and positive predictive values for rebleeding (all patients).

	Sensitivity (%)	Specificity (%)	Positive Predictive Value (%)	Negative Predictive Value (%)
Endoscopist judgment	82.5	60.9	19	96.9
Rockall Score > 2	90	33.8	13.1	96.8
Rockall Score >4	75	66.2	19.7	96
Cedars–Sinai Score > 2	90	33.8	13.1	96.8
Cedars–Sinai Score >4	62.5	75.1	21.7	94.8
Blatchford Score >0	100	0.3	10	100
Blatchford Score >11	72.5	58.4	16.2	95.0
Baylor Score >10	50	71.4	16.3	92.8

**Table 4 jcm-09-00408-t004:** Sensitivity, specificity and positive predictive values for mortality (endoscopic treatment group).

	Sensitivity (%)	Specificity (%)	Positive Predictive Value (%)	Negative Predictive Value (%)
Endoscopist judgment	100	65	12.8	100
Rockall Score > 2	100	5.6	5.2	100
Rockall Score >4	90.9	39.3	7.1	98.8
Cedars–Sinai Score > 2	85	47.6	15.2	96.6
Cedars–Sinai Score >4	72.7	53.3	7.4	97.4
Blatchford Score >0	100	0	4.9	--
Blatchford Score >11	90.9	47.2	8.1	99.0
Baylor Score >10	63.6	55.2	6.7	96.8

**Table 5 jcm-09-00408-t005:** Sensitivity, specificity and positive predictive values for rebleeding (endoscopic treatment group).

	Sensitivity (%)	Specificity (%)	Positive Predictive Value (%)	Negative Predictive Value (%)
Endoscopist judgment	88.9	32.8	20.1	93.9
Rockall Score > 2	94.4	5.3	16	83.3
Rockall Score >4	80.6	41.3	20.7	91.8
Cedars–Sinai Score > 2	91.7	18	17.6	91.9
Cedars–Sinai Score >4	66.7	55.6	22.2	89.7
Blatchford Score >0	100	0	16	--
Blatchford Score >11	72.2	48.7	21.1	90.2
Baylor Score >10	48.7	55	18.3	83.9

**Table 6 jcm-09-00408-t006:** Sensitivity, specificity, positive and negative predictive values for predicting rebleeding and mortality according to the experience of the endoscopists.

	Sensitivity (%)	Specificity (%)	Positive Predictive Value (%)	Negative Predictive Value (%)
*Rebleeding*				
Less experienced	77.8	65.6	13.5	97.7
More experienced	83.9	58.2	21.3	96.4
*Mortality*				
Less experienced	100	79.7	6.7	100
More experienced	100	75.2	15.1	100

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
