# Peer review of "Endoscopist’s Judgment Is as Useful as Risk Scores for Predicting Outcome in Peptic Ulcer Bleeding: A Multicenter Study"

_jcm, 2020, doi:10.3390/jcm9020408_

Round 1

Reviewer 1 Report

The comparison was between the endoscopist's judgement and the scores from established criteria. As such, was an initial comparison made between the four different risk-scoring systems (Rockall, Glasgow-Blatchford, Baylor and the Cedars-Sinai scores)? Why was it assumed that "high" and/or "Intermediate" (Rockall, and Cedars-Sinai) and "high" (Glasgow-Blatchford, and Baylor) all were at the same level of prediction of the pimary outcomes?  For instance why not just "high" in all four?  What was the rationale and how was it supported? We are told the analysis was repeated using thresholds of the Rockall and Cedars-Sinai scores greater than 4 (this would allow just "high" for these two,) and the Glasgow Blatchford score greater than 11.  This seems arbitrary.Is there an explanation for choosing 11, and not 8 or 13?   And finally, what happened to the Baylor score?  Did it not have an alternative adjusted cut-off point. Is there an explanation?  Was a power analysis performed to determine the 'n' needed in each group? We are told 401 patients were recruited, but on what basis? The calculation for specificity and sensitivity and also positve and negative predictive values requires a "gold standard" set of criteria against which the unknown or uncertain set of critera are compared. In this case it appears that neither endoscopist's judgement nor the scores were clearly a "gold standard". Critera. Is it correct to assume that the comparisons were made against the actual clinical outcomes of rebleeding and mortality? It may be helpful to make this clear. (It would help if ALL percentages in Table 1 were shown in parentheses next to eact frequency.  In the manuscript, some are present and some missing. Figures 1 and 2 … For these comprisons, it is not clear whether the original cut-off ponts or the alternative cut-off points were used for the three scoring criteria (Bayliss did not have an alternative cut-off). This is made clear in Tables 2,3,4,5, but not Figures 1 and 2 .   

Line 95….  Please confirm the use of "reposition"

Line 76 …     "…with the of the…"

Line 70….   Appears to need re-wording

Author Response

Reviewer 1.

Thanks for the useful comments.

The comparison was between the endoscopist's judgement and the scores from established criteria. As such, was an initial comparison made between the four different risk-scoring systems (Rockall, Glasgow-Blatchford, Baylor and the Cedars-Sinai scores)?

Comparison between scores was not the primary objective of the study. Anyway, comparison between scores was done, although figures show only p values for the comparison between endoscopists judgement ant the scores. It has been added to the text that scores did not differ between them (Lines 176-178).

Why was it assumed that "high" and/or "Intermediate" (Rockall, and Cedars-Sinai) and "high" (Glasgow-Blatchford, and Baylor) all were at the same level of prediction of the primary outcomes?  For instance, why not just "high" in all four? What was the rationale and how was it supported? We are told the analysis was repeated using thresholds of the Rockall and Cedars-Sinai scores greater than 4 (this would allow just "high" for these two,) and the Glasgow Blatchford score greater than 11. This seems arbitrary. Is there an explanation for choosing 11, and not 8 or 13? And finally, what happened to the Baylor score? Did it not have an alternative adjusted cut-off point. Is there an explanation?

Global performance of prognostic methods – including the scores- was evaluated by using the Area Under the ROC curve (AUROC). AUROC evaluation is independent of the cut-off used to calculate sensitivity and specificity. So, the cut-off selected did not affected the evaluation of the diagnostic accuracy of each method.

Regarding to the cut-off used to determine sensitivity and specificity for each score, we selected the cut-off stablished in the original articles (references 11 to 14) as stated in the text (Lines 122-123). As most of the scores are widely used, we considered that it was unreasonable to calculate the particular cut-off performing best in our particular sample. When more than a cut-off was provided in the original study, (as, for example, in the case of Rockall score) we provided sensitivity and specificity for each cut-off.

Was a power analysis performed to determine the 'n' needed in each group? We are told 401 patients were recruited, but on what basis?

A sample size of 400 patients was anticipated for an estimated rate of rebleeding of 10%, a 10-percentage point wide confidence interval and expecting a sensitivity of 80% and a specificity of 80%. This has been added to the text (Lines 141-143).

Additionally, as formal sample size calculation is difficult for this kind of studies (Karimollah, HT, Journal of Biomedical Informatics 2014) the sample size was set in the range of most of the previously validation studies.

The calculation for specificity and sensitivity and also positive and negative predictive values requires a "gold standard" set of criteria against which the unknown or uncertain set of criteria are compared. In this case it appears that neither endoscopist's judgement nor the scores were clearly a "gold standard" criteria. Is it correct to assume that the comparisons were made against the actual clinical outcomes of rebleeding and mortality? It may be helpful to make this clear.

In this case, neither the scores nor clinical judgement were used as Gold Standard. As the reviewer correctly states, gold-standard was established according patients’ outcomes during clinical follow-up. This has been now clearly stated in the text (Line 112).

It would help if ALL percentages in Table 1 were shown in parentheses next to each frequency.  In the manuscript, some are present and some missing.

Done.

Figures 1 and 2. For these comparisons, it is not clear whether the original cut-off points or the alternative cut-off points were used for the three scoring criteria (Bayliss did not have an alternative cut-off). This is made clear in Tables 2,3,4,5, but not Figures 1 and 2.

Global performance of prognostic methods – including the scores- was evaluated by using the Area Under the ROC curve (AUROC). AUROC evaluation of the performance of a diagnostic method neither requires nor uses cut-off values. A cut-off is only necessary to determine sensitivity and specificity but did not affected the evaluation of the diagnostic accuracy by AUROC. For this reason, cut-offs are clearly stated in the tables but not in the AUROC figures.

Line 95….  Please confirm the use of "reposition"

Corrected, thank you.

Line 76 …     "…with the of the…"

Corrected, thank you

Line 70….   Appears to need re-wording

Done

Reviewer 2 Report

Dear authors,

Thank you for submitting this manuscript. This thoughtful study raises the question of whether clinical judgment is as accurate as scoring systems in predicting bleeding and mortality in peptic ulcer disease. In an era of prediction models for bleeding and mortality, clinical judgment is important but very difficult to study due to the heterogeneity of clinical judgment. 

The authors are correct in that current scoring systems do not capture certain factors such as whether providers felt that adequate endoscopic hemostasis was achieved and were derived from different causes of gastrointestinal hemorrhage.

This is a well-written manuscript on a clinically important topic. There are a few comments and suggestions that I hope will enhance the quality of this study and manuscript.

Methods

1. More information about the degree of training among clinicians would be helpful. The authors note that endoscopists had 5 or more years of experience in therapeutic endoscopy.  What was the mean and range of years of experience among endoscopists? 

2. More details regarding the endoscopist's assessment of risk of mortality and bleeding would be helpful. For example, was the risk of mortality and bleeding assessed in a dichotomous fashion (high risk versus low risk) or was there a scoring system used to indicate the endoscopist's assessment such as a Likert scale for a variety of variables (amount of blood loss during the procedure, Forrest classification).

3. Given the subjective nature of clinical judgment, it would be helpful to understand what factors these endoscopists used to judge the mortality and bleeding risk. Did the investigators seek to understand what factors contributed to their assessment of risk of bleeding or death? 

4. The study was conducted several years ago prior to the AIMS65 score, which was found to be most predictive of mortality in several studies comparing multiple bleeding risk scores (Stanley et al. BMJ 2017; Robertson et al. Gastrointest Endoscopy 2016). It would have been nice to see the authors include the AIMS65 score in their study. 

5. Was thermal therapy such as heater probe or bipolar electrocautery not available during the study period? The authors indicate that hemostatic clips and injections were used, but thermal therapy is another acceptable method of achieving hemostasis. 

6. How was the endoscopist's subjective assessment standardized among different centers? Was there a dedicated survey or scale? Because this was a multicenter study, it would be helpful to understand if there was a standardized method such as a form or scale to record or inform the endoscopist's assessment.

7. The authors should indicate the threshold of significance (p-value) used in their analysis for the Bonferroni correction.

Results

1. Did years of experience affect the accuracy of clinical judgment in comparison to risk scores? In the conclusion, the authors note that a post-hoc analysis did not show differences in the predictive value of clinical judgment by endoscopist experience. It would strengthen the manuscript if the authors would show this analysis based on years of experience and accuracy of clinical judgment as supplemental data. 

2. In table 1, the authors indicate that 10 patients in the endoscopic treatment group but 11 patients in the whole group underwent surgery. In the methods section, the authors note that surgery was performed in patients who experienced persistent bleeding after a second endoscopic treatment. Can the authors account for the discrepancy between the number of patients who underwent surgery in the endoscopic treatment group and the whole group? Is this an error, or is there a patient who underwent surgery without endoscopic treatment? 

3. The authors note that prediction from endoscopist judgment and risk scores for the risk of rebleeding and death were low and moderate with low specificity and low positive predictive value. This is interesting yet not surprising. There are new studies seeking to apply machine learning technology to try to better predict need for endoscopic intervention or death (Shung et al. Gastroenterology 2020). 

4. Do the authors have p-values or some method of demonstrating the statistical difference between sensitivity, specificity, positive predictive value, and negative predictive value for tables 2-5? 

Author Response

Dear authors,

Thank you for submitting this manuscript. This thoughtful study raises the question of whether clinical judgment is as accurate as scoring systems in predicting bleeding and mortality in peptic ulcer disease. In an era of prediction models for bleeding and mortality, clinical judgment is important but very difficult to study due to the heterogeneity of clinical judgment.

The authors are correct in that current scoring systems do not capture certain factors such as whether providers felt that adequate endoscopic hemostasis was achieved and were derived from different causes of gastrointestinal hemorrhage.

This is a well-written manuscript on a clinically important topic. There are a few comments and suggestions that I hope will enhance the quality of this study and manuscript.

Thank you.

Methods

  1. More information about the degree of training among clinicians would be helpful. The authors note that endoscopists had 5 or more years of experience in therapeutic endoscopy. What was the mean and range of years of experience among endoscopists?

Mean years of experience in gastrointestinal endoscopy were 14.1 with a range from 8 to 20.

  1. More details regarding the endoscopist's assessment of risk of mortality and bleeding would be helpful. For example, was the risk of mortality and bleeding assessed in a dichotomous fashion (high risk versus low risk) or was there a scoring system used to indicate the endoscopist's assessment such as a Likert scale for a variety of variables (amount of blood loss during the procedure, Forrest classification).

The endoscopist classified the patients as high-risk group and low-risk group both for rebleeding and mortality. This specification has been included in the text. (Line 116)

  1. Given the subjective nature of clinical judgment, it would be helpful to understand what factors these endoscopists used to judge the mortality and bleeding risk. Did the investigators seek to understand what factors contributed to their assessment of risk of bleeding or death?

No. No guideline was provided to the endoscopists on how to classify the patients. It is reasonable to think that comorbidity, the severity of the bleeding, the type of endoscopic lesion and the subjective adequacy of the treatment, were considered to classify the patients. However, to avoid influencing the subjective evaluation process no attempt was made to individually measure these factors.

  1. The study was conducted several years ago prior to the AIMS65 score, which was found to be most predictive of mortality in several studies comparing multiple bleeding risk scores (Stanley et al. BMJ 2017; Robertson et al. Gastrointest Endoscopy 2016). It would have been nice to see the authors include the AIMS65 score in their study.

We agree. Unfortunately, we do not have these data. We have added this point as a possible further development of our study.

  1. Was thermal therapy such as heater probe or bipolar electrocautery not available during the study period? The authors indicate that hemostatic clips and injections were used, but thermal therapy is another acceptable method of achieving hemostasis.

In our setting, thermal therapy is rarely used, and it is not available in most centers. When the study was performed standard hemostatic methods were dual injection with adrenaline plus a sclerosant or adrenaline plus clip.

  1. How was the endoscopist's subjective assessment standardized among different centers? Was there a dedicated survey or scale? Because this was a multicenter study, it would be helpful to understand if there was a standardized method such as a form or scale to record or inform the endoscopist's assessment.

We assumed from the beginning that endoscopist judgement was a subjective evaluation. In consequence, no guideline was provided to avoid interference with this subjective evaluation. Any attempt to standardize would have interfered with what was planned to be evaluated. This information has been added to the methods.  (Lines 116-117)

  1. The authors should indicate the threshold of significance (p-value) used in their analysis for the Bonferroni correction.

This information has been added to the text. (Line 140).

Results

  1. Did years of experience affect the accuracy of clinical judgment in comparison to risk scores? In the conclusion, the authors note that a post-hoc analysis did not show differences in the predictive value of clinical judgment by endoscopist experience. It would strengthen the manuscript if the authors would show this analysis based on years of experience and accuracy of clinical judgment as supplemental data.

The analysis was performed post-hoc, and no significant differences were found between experienced an unexperienced endoscopists. We have added these data to the text (Lines 199-209)

  1. In table 1, the authors indicate that 10 patients in the endoscopic treatment group but 11 patients in the whole group underwent surgery. In the methods section, the authors note that surgery was performed in patients who experienced persistent bleeding after a second endoscopic treatment. Can the authors account for the discrepancy between the number of patients who underwent surgery in the endoscopic treatment group and the whole group? Is this an error, or is there a patient who underwent surgery without endoscopic treatment?

One patient classified -probably misclassified- as low risk by the first endoscopy presented a severe rebleeding that could not be controlled endoscopically, thus requiring surgery. Numbers are correct.

  1. The authors note that prediction from endoscopist judgment and risk scores for the risk of rebleeding and death were low and moderate with low specificity and low positive predictive value. This is interesting yet not surprising. There are new studies seeking to apply machine learning technology to try to better predict need for endoscopic intervention or death (Shung et al. Gastroenterology 2020).

A very interesting work. We have added a comment on the article in the text.

  1. Do the authors have p-values or some method of demonstrating the statistical difference between sensitivity, specificity, positive predictive value, and negative predictive value for tables 2-5?

Comparison between diagnostic methods are better evaluated by comparing the differences in the areas under the ROC curves. The differences between sensitivities and specificities between scores are highly dependent on the cut-off selected. Giving p values for each comparison would give the false perception that scores differ between them. One to one comparison for each value given in tables 2-5 was not attempted.